# Empirical Phase Diagram for Three-layer Neural Networks with Infinite Width

**Hanxu Zhou**[1]**, Qixuan Zhou**[1]**, Zhenyuan Jin**[1]**, Tao Luo**[1,2]**, Yaoyu Zhang**[1,3]**, Zhi-Qin John Xu**[1]*

[1] School of Mathematical Sciences, Institute of Natural Sciences, MOE-LSC
and Qing Yuan Research Institute, Shanghai Jiao Tong University
[2] CMA-Shanghai, Shanghai Artificial Intelligence Laboratory
[3] Shanghai Center for Brain Science and Brain-Inspired Technology

## Abstract

Substantial work indicates that the dynamics of neural networks (NNs) is closely related to their initialization of parameters. Inspired by the phase diagram for two-layer ReLU NNs with infinite width (Luo et al., 2021), we make a step towards drawing a phase diagram for three-layer ReLU NNs with infinite width. First, we derive a normalized gradient flow for three-layer ReLU NNs and obtain two key independent quantities to distinguish different dynamical regimes for common initialization methods. With carefully designed experiments and a large computation cost, for both synthetic datasets and real datasets, we find that the dynamics of each layer also could be divided into a linear regime and a condensed regime, separated by a critical regime. The criteria is the relative change of input weights (the input weight of a hidden neuron consists of the weight from its input layer to the hidden neuron and its bias term) as the width approaches infinity during the training, which tends to $0$, $+\infty$ and $O(1)$, respectively. In addition, we also demonstrate that different layers can lie in different dynamical regimes in a training process within a deep NN. In the condensed regime, we also observe the condensation of weights in isolated orientations with low complexity. Through experiments under three-layer condition, our phase diagram suggests a complicated dynamical regimes consisting of three possible regimes, together with their mixture, for deep NNs and provides a guidance for studying deep NNs in different initialization regimes, which reveals the possibility of completely different dynamics emerging within a deep NN for its different layers.

## 1 Introduction

As neural networks (NNs) have been employed to extensive practical and scientific problems, such as computer vision, natural language processing, and automatic driving, etc. Studying the rationale behind them can help us comprehend the principles behind NNs and better use NNs. For example, the initialization of neural network parameters plays an important role in the training dynamics and generalization of NN fitting.

Several types of initialization schemes are often used in practical trainings, such as He initialization (He et al., 2015), LeCun initialization (LeCun et al., 2012), Xavier initialization (Glorot and Bengio, 2010), etc. Theoretical study further explores how initialization affects the training dynamics of NNs. For example, the NTK scaling (Jacot et al., 2018; Chizat and Bach, 2019; Arora et al., 2019; Zhang et al., 2020) leads to a training where parameters stay at a small neighborhood of the initialization, therefore, the NN after training can be well approximated by the first-order Taylor expansion at the

---

*Corresponding author: xuzhiqin@sjtu.edu.cn.

36th Conference on Neural Information Processing Systems (NeurIPS 2022).

initial point, i.e., linear dynamics. The mean-field scaling (Mei et al., 2019; Rotskoff and Vanden-Eijnden, 2018; Chizat and Bach, 2018; Sirignano and Spiliopoulos, 2020) leads to a training where parameters significantly deviates from the initialization, therefore, the NN dynamics exhibits clearly non-linear behavior. Luo et al. (2021) systematically study the effect of initialization for two-layer ReLU NN with infinite width by establishing a phase diagram, which shows three distinct regimes, i.e., linear regime, critical regime and condensed regime, based on the relative change of input weights (the input weight of a hidden neuron consists of the weight from its input layer to the hidden neuron and its bias term) as the width approaches infinity during the training, which tends to $0$, $O(1)$ and $+\infty$, respectively. The condensed regime is named so, because empirical study finds that orientations of the input weights condense in several isolated orientations, which is a strong feature learning behavior, an important characteristic of deep learning.

The study in Luo et al. (2021) is limited in two-layer NNs, however, empirical studies show that deep networks can learn faster and generalize better than shallow networks in both real data and synthetic data (He et al., 2016; Arora et al., 2018; Xu and Zhou, 2021). Because of multi-layer structure and non-linearity, the study of multi-layer nonlinear NNs is often challenging. For example, the mean-field theory is established on two-layer NNs and difficult to be extended to multi-layer NNs (Mei et al., 2019; Rotskoff and Vanden-Eijnden, 2018; Chizat and Bach, 2018; Sirignano and Spiliopoulos, 2020). Even for empirical study, excessive hyper-parameters often set an obstacle to obtain clear phenomena as well, where multi-layer NNs could have distinct characteristics compared with shallow two-layer NNs. For example, due to the existence of other layers, how is the dynamical regime of the weights of a layer in multi-layer NNs different from two-layer ones and can weights of two different layers experience distinct dynamical behaviors in one training process? Answering these questions is important to understand how the dynamics of multi-layer NNs depend on the initial hyper-parameters, thus, leading to a deeper understanding of deep NNs and providing guidance for tuning parameters in practical training a tool for the theory community. For instance, if we want to study the nonlinear effect of a hidden layer, we can set this interested hidden layer in the non-linear regime and all other hidden layers in the linear regime. Further, we can continue to study how the non-linear effect of a hidden layer depends on its depth.

Three-layer network is a good starting point to study multi-layer NNs as it only has two hidden layers for studying the interaction between layers. Besides, to have a clear boundary of different dynamical regimes, we focus on three-layer ReLU NNs with large-width limit. A difficulty to overcome is to find appropriate quantities required for describing a phase diagram, which should be able to identify distinct different regimes, and can also account for the same dynamical behavior when hyper-parameters are different but those quantities are the same. To this end, we derive a normalized gradient flow for a three-layer ReLU NN. Since we only care about the training trajectory up to a time scaling, we obtain three independent quantities that meet our requirements. And then, we further simplify the setting by assuming weights of the two outer layers use a same initialization scheme as the commonly used in practice, and we obtain two key independent quantities. With carefully designed experiments and a large computation cost, for both synthetic datasets and real datasets, we find that weights of each layer also have a linear regime and a condensed regime, separated by a critical regime. The criteria is the relative change of input weights as the width approaches infinity during the training, which tends to $0$, $+\infty$ and $O(1)$, respectively. In addition, we also find that different layers can lie in different dynamical regimes in a training process. In the condensed regime, we similarly observe the condensation of weights in isolated orientations.

This work empirically draws a phase diagram for three-layer ReLU NNs with large width, as shown in Fig. 1. For deeper networks, this work suggests that the dynamical regime for the weights of each layer is similar to that of a two-layer NN and different layers can lie in distinct regimes in a training process. For NNs with not too large width, the boundary between regimes is not sharp but this work still sheds lights on how the initialization affects the dynamics of weights in different layers. Our study on phase diagram of three-layer ReLU NNs thus lays an important foundation for future work on the dynamical behavior of multi-layer NNs and corresponding implicit regularizations at each of the identified regimes and guides the parameter initialization tuning in the practice.

## 2 Related works

A series of works have shown that global minima can be found for NNs with several specific initializations in linear regime (Zou et al., 2018; Allen-Zhu et al., 2019; Du et al., 2019; E et al., 2020) or

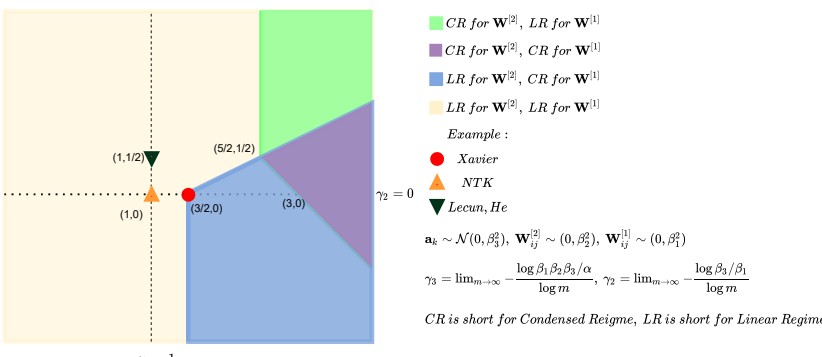

Figure 1: Phase diagram of three-layer ReLU NNs at infinite-width limit. The four regimes are empirically partitioned, and the marked examples are studied in existing literature (see Table 1 for details).

for the two-layer ReLU NNs in the whole linear regime (Luo et al., 2021). Geiger et al. (2020) probe the dynamic crossover between the NTK limit and the mean-feild limit, and Yang (2020); Yang and Littwin (2021) show that tangent kernel of a randomly initialized neural network with 'arbitrary architecture' converges to a deterministic limit, as the network width goes to infinity. Chen et al. (2021) prove that for wide shallow NNs under the mean-field scaling and with a general class of activation functions, when the input dimension is at least the size of the training set, the training loss converges to zero at a linear rate under gradient flow.

In the condensation regime, a large network with condensation is effectively a network of only a few effective neurons, leading to an output function with low complexity (Bartlett and Mendelson, 2002), thus, may provide a possible explanation for good generalization performance of large NNs (Breiman, 1995; Zhang et al., 2021). The condensation suggests a bias towards simple function, which is consistent with the frequency principle that NNs tend to learn data by low-frequency function (Xu et al., 2019; Rahaman et al., 2019; Xu et al., 2020, 2022). NNs have significant feature learning in the condensation regimes. To actively learn features, Lyu et al. (2021) use small initialization to study how gradient descent can lead a NN to a solution of 'max-margin' solution.

How the condensation happens during the initial training stage is explored for ReLU activation function (Maennel et al., 2018; Pellegrini and Biroli, 2020) and general differential activation functions (Zhou et al., 2021). The embedding principle (Zhang et al., 2021a,b) shows that the loss landscape of an NN contains all critical points/functions of all the narrower NNs, thus providing a basis for why a large network can condense during the training.

## 3  Preliminary

For convenience, we consider a three-layer NN with $m$ hidden neurons for each layer,

$$f_{\boldsymbol{\theta}}(\boldsymbol{x}) = \frac{1}{\alpha}\boldsymbol{a}^T \sigma(\boldsymbol{W}^{[2]}\sigma(\boldsymbol{W}^{[1]}\boldsymbol{x})), \tag{1}$$

where $\boldsymbol{x} \in \mathbb{R}^d$, $\alpha$ is the scaling factor, $m$ is the width of both layers, $\boldsymbol{\theta} = \text{vec}\{\boldsymbol{a}, \boldsymbol{W}^{[2]}, \boldsymbol{W}^{[1]}\}$ with $\boldsymbol{a} \in \mathbb{R}^m$, $\boldsymbol{W}^{[2]} \in \mathbb{R}^{m \times m}$, $\boldsymbol{W}^{[1]} \in \mathbb{R}^{m \times d}$ is the set of parameters initialized by $\boldsymbol{a}_k \sim \mathcal{N}(0, \beta_3^2)$, $\boldsymbol{W}_{kk'}^{[2]} \sim \mathcal{N}(0, \beta_2^2)$, $\boldsymbol{W}_{kk'}^{[1]} \sim \mathcal{N}(0, \beta_1^2)$, i.i.d.. The bias term $b_k^{[1]}$ can be incorporated by expanding $\boldsymbol{x}$ and $\boldsymbol{W}^{[1]}$ to $(\boldsymbol{x}^\mathsf{T}, 1)^\mathsf{T}$ and $(\boldsymbol{W}^{[1]}, b_k^{[1]})^\mathsf{T}$, while $b_k^{[2]}$ can be incorporated by expanding $\bar{\boldsymbol{x}} = \sigma(\boldsymbol{W}^{[1]}\boldsymbol{x})$ and $\boldsymbol{W}^{[2]}$ to $(\bar{\boldsymbol{x}}^\mathsf{T}, 1)^\mathsf{T}$ and $(\boldsymbol{W}^{[2]}, b_k^{[2]})^\mathsf{T}$.

At the infinite-width limit $m \to \infty$, given $\beta_1, \beta_2, \beta_3 \sim O(1)$, for $\alpha \sim \sqrt{m_1 m_2}$, the gradient flow of NNs can be approximated by a linear dynamics of neural tangent kernel (NTK) (Jacot et al. (2018)), whereas given $\beta_1 \sim \sqrt{\frac{2}{m_1+d}}, \beta_2 \sim \sqrt{\frac{2}{m_1+m_2}}, \beta_3 \sim \sqrt{\frac{2}{m_2+1}}$, for $\alpha \sim O(1)$ gradient flow of NN exhibits highly nonlinear dynamics of Xavier (Glorot and Bengio (2010)).

**Assumption 3.1.** For convenience, we assume that the number of neurons in each hidden layer is equal, i.e. $m_1 = m_2 = m$.

Since the output is linear with respect to $\boldsymbol{a}$, $\boldsymbol{a}$ is not the main concern, and the initialization methods used for a neural network in practice often has $\beta_2 = B\beta_3$, where $B$ is a constant in dependent of $m$. So, we take the following assumption.

**Assumption 3.2.** $\beta_2 = B\beta_3$, where $B$ is a constant independent of $m$.

**Cosine similarity:** The cosine similarity for two vectors $\boldsymbol{u}$ and $\boldsymbol{v}$ is defined as

$$D(\boldsymbol{u}, \boldsymbol{v}) = \frac{\boldsymbol{u}^\mathsf{T}\boldsymbol{v}}{(\boldsymbol{u}^\mathsf{T}\boldsymbol{u})^{1/2}(\boldsymbol{v}^\mathsf{T}\boldsymbol{v})^{1/2}}. \tag{2}$$

## 4 Rescaling and the normalized model of the three layer neural network

To identify sharply distinctive regimes/states, we take the infinite-width limit $m \to \infty$ as our starting point. Identification of the coordinates is important for drawing the phase diagram. There are some guiding principles for finding the coordinates of a phase diagram (Luo et al., 2021): (i) They should be effectively independent; (ii) Given a specific coordinate in the phase diagram, the learning dynamics of all the corresponding NNs statistically should be similar up to a time scaling; (iii) They should well differentiate dynamical differences except for the time scaling.

Guided by above principles, in this section, we perform the following rescaling procedure for a fair comparison between different choices of hyperparameters and obtain a normalized model with two independent quantities irrespective of the time scaling of the gradient flow dynamics.

The empirical risk is

$$R_S(\boldsymbol{\theta}) = \frac{1}{2n}\sum_{i=1}^{n}(f_{\boldsymbol{\theta}}(\boldsymbol{x}_i) - y_i)^2. \tag{3}$$

Then the training dynamics based on gradient descent (GD) at the continuous limit obeys the following gradient flow of $\boldsymbol{\theta}$,

$$\frac{\mathrm{d}\boldsymbol{\theta}}{\mathrm{d}t} = -\nabla_{\boldsymbol{\theta}}R_S(\boldsymbol{\theta}). \tag{4}$$

More precisely, we obtain that:

$$\frac{\mathrm{d}\boldsymbol{a}}{\mathrm{d}t} = -\frac{1}{n}\sum_{i=1}^{n}\frac{1}{\alpha}\sigma(\boldsymbol{W}^{[2]}\sigma(\boldsymbol{W}^{[1]}\boldsymbol{x}_i))e_i,$$

$$\frac{\mathrm{d}\boldsymbol{W}^{[2]}}{\mathrm{d}t} = -\frac{1}{n}\sum_{i=1}^{n}\frac{1}{\alpha}\boldsymbol{a}\odot\sigma'(\boldsymbol{W}^{[2]}\sigma(\boldsymbol{W}^{[1]}\boldsymbol{x}_i))\sigma(\boldsymbol{W}^{[1]}\boldsymbol{x}_i)^\mathsf{T}e_i, \tag{5}$$

$$\frac{\mathrm{d}\boldsymbol{W}^{[1]}}{\mathrm{d}t} = -\frac{1}{n}\sum_{i=1}^{n}\frac{1}{\alpha}\boldsymbol{W}^{[2]\mathsf{T}}(\boldsymbol{a}\odot\sigma'(\boldsymbol{W}^{[2]}\sigma(\boldsymbol{W}^{[1]}\boldsymbol{x}_i)))\odot\sigma'(\boldsymbol{W}^{[1]}\boldsymbol{x}_i)\boldsymbol{x}_i^\mathsf{T}e_i,$$

where $e_i = \left(\frac{1}{\alpha}\boldsymbol{a}^\mathsf{T}\sigma(\boldsymbol{W}^{[2]}\sigma(\boldsymbol{W}^{[1]}\boldsymbol{x}_i)) - y_i\right)$, the operation $\odot$ is the Hadamard product. Then, setting

$$\overline{\boldsymbol{a}} = \frac{1}{\beta_3}\boldsymbol{a}, \ \overline{\boldsymbol{W}}^{[2]} = \frac{1}{\beta_2}\boldsymbol{W}^{[2]}, \ \overline{\boldsymbol{W}}^{[1]} = \frac{1}{\beta_1}\boldsymbol{W}^{[1]}, \tag{6}$$

so that all the parameters are picked from standard normal distribution. By the chain rule of derivative and the homogeneity of ReLU, i.e. $\sigma(a\boldsymbol{u}) = a\sigma(\boldsymbol{u})$ and $\sigma'(a\boldsymbol{u}) = \sigma'(\boldsymbol{u})$, where $a > 0$ is a scalar and $\boldsymbol{u}$ is a vector, we obtain that:

Table 1: Common initialization methods with their scaling parameters

| Name | $\alpha$ | $\boldsymbol{a}$ | $\boldsymbol{W}^{[2]}$ | $\boldsymbol{W}^{[1]}$ | $\kappa_2$ | $\kappa_3$ | $\gamma_2$ | $\gamma_3$ |
|---|---|---|---|---|---|---|---|---|
| NTK
Jacot et al. (2018) | $\sqrt{m_1 m_2}$ | $1$ | $1$ | $1$ | $1$ | $\sqrt{\dfrac{1}{m_1 m_2}}$ | $0$ | $1$ |
| Lecun
LeCun et al. (2012) | $1$ | $\sqrt{\dfrac{1}{m_2}}$ | $\sqrt{\dfrac{1}{m_1}}$ | $\sqrt{\dfrac{1}{d}}$ | $\sqrt{\dfrac{d}{m_2}}$ | $\sqrt{\dfrac{1}{m_1 m_2 d}}$ | $\dfrac{1}{2}$ | $1$ |
| He
He et al. (2015) | $1$ | $\sqrt{\dfrac{2}{m_2}}$ | $\sqrt{\dfrac{2}{m_1}}$ | $\sqrt{\dfrac{2}{d}}$ | $\sqrt{\dfrac{d}{m_2}}$ | $\sqrt{\dfrac{8}{m_1 m_2 d}}$ | $\dfrac{1}{2}$ | $1$ |
| Xavier
Glorot and Bengio (2010) | $1$ | $\sqrt{\dfrac{2}{m_2+1}}$ | $\sqrt{\dfrac{2}{m_1+m_2}}$ | $\sqrt{\dfrac{2}{d+m_1}}$ | $\sqrt{\dfrac{d+m_1}{m_2+1}}$ | $\sqrt{\dfrac{8/(m_1+m_2)}{(m_2+1)(d+m_1)}}$ | $0$ | $\dfrac{3}{2}$ |

$$
\frac{\mathrm{d}\overline{\boldsymbol{a}}}{\mathrm{d}\overline{t}} = -\left( \frac{1}{n} \sum_{i=1}^{n} \kappa_3 \sigma(\overline{\boldsymbol{W}}^{[2]} \sigma(\overline{\boldsymbol{W}}^{[1]} \boldsymbol{x}_i)) \right) e_i,
$$

$$
\frac{\mathrm{d}\overline{\boldsymbol{W}}^{[2]}}{\mathrm{d}\overline{t}} = -\kappa_1^2 \left( \frac{1}{n} \sum_{i=1}^{n} \kappa_3 \overline{\boldsymbol{a}} \odot \sigma'(\overline{\boldsymbol{W}}^{[2]} \sigma(\overline{\boldsymbol{W}}^{[1]} \boldsymbol{x}_i)) \sigma(\overline{\boldsymbol{W}}^{[1]} \boldsymbol{x}_i)^{\mathbf{T}} \right) e_i, \tag{7}
$$

$$
\frac{\mathrm{d}\overline{\boldsymbol{W}}^{[1]}}{\mathrm{d}\overline{t}} = -\kappa_2^2 \left( \frac{1}{n} \sum_{i=1}^{n} \kappa_3 \overline{\boldsymbol{W}}^{[2]\mathbf{T}} (\overline{\boldsymbol{a}} \odot \sigma'(\overline{\boldsymbol{W}}^{[2]} \sigma(\overline{\boldsymbol{W}}^{[1]} \boldsymbol{x}_i))) \odot \sigma'(\overline{\boldsymbol{W}}^{[1]} \boldsymbol{x}_i) \boldsymbol{x}_i^{\mathbf{T}} \right) e_i,
$$

where, we introduce four scaling parameters

$$
\kappa_1 = \frac{\beta_3}{\beta_2}, \; \kappa_2 = \frac{\beta_3}{\beta_1}, \; \kappa_3 = \frac{\beta_1 \beta_2 \beta_3}{\alpha}, \; \overline{t} = (\alpha \Pi_{i=1}^{3} \kappa_i)^{-\frac{2}{3}} t. \tag{8}
$$

Note that $\kappa_1$, $\kappa_2$ and $\kappa_3$ do not follow principle (ii) and (iii) above at infinite-width limit. They are in general functions of $m$, which attains $0, O(1), +\infty$ at $m \to \infty$. To account for such dynamical difference under different widely considered power-law scalings of $\alpha, \beta_1, \beta_2$ and $\beta_3$ shown in Table 1 and based on our assumption 3.1 and 3.2, we arrive at

$$
\gamma_2 = \lim_{m \to \infty} -\frac{\log \kappa_2}{\log m}, \gamma_3 = \lim_{m \to \infty} -\frac{\log \kappa_3}{\log m}, \tag{9}
$$

where our assumptions 3.1 and 3.2 mean $\kappa_1 = B$, leading to $\gamma_1 = \lim_{m \to \infty} -\frac{\log \kappa_1}{\log m} = 0$. So we discuss the phase diagram of three layer network under the coordinates: $(\gamma_1, \gamma_2, \gamma_3) = (0, \gamma_2, \gamma_3)$, and focus on $(\gamma_2, \gamma_3)$.

*Remark* 4.1. Here we list some commonly-used initialization methods and/or related works with their scaling parameters as shown in Table 1.

## 5 Empirical phase diagram

### 5.1 Experimental setup

Throughout this section, we use three-layer fully-connected neural networks with size, $d$-$m$-$m$-$d_{\text{out}}$. The input dimension $d$ is determined by the training data, i.e., $d = 1$ for synthetic data and $d = 28 \times 28$ for MNIST. The output dimension is $d_{\text{out}} = 1$ for synthetic data and $d_{\text{out}} = 10$ for MNIST. The number of hidden neurons $m$ is specified in each experiment. All parameters are initialized by a Gaussian distribution $N(0, \text{var})$, where var depends on $\beta_1, \beta_2$ and $\beta_3$. The total data size is $n$. The training method is gradient descent with full batch, learning rate lr and MSE loss. For synthetic data, we use a simple 1-d problem of 4 training points.

## 5.2 Intuitive experiments of synthetic data

With $\gamma_3$ and $\gamma_2$ as coordinates, in this subsection, we illustrate through experiments the behavior of a diversity of typical cases over the phase diagram regarding $\boldsymbol{W}^{[1]}$ and $\boldsymbol{W}^{[2]}$ respectively. We use a simple 1-d problem of 4 training points, which allows an easy visualization.

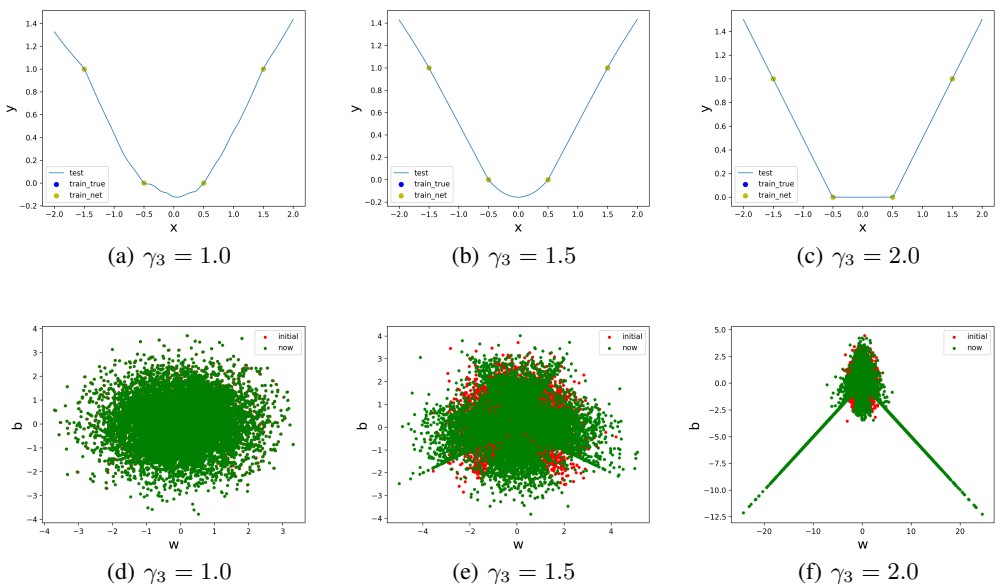

(a) $\gamma_3 = 1.0$        (b) $\gamma_3 = 1.5$        (c) $\gamma_3 = 2.0$

(d) $\gamma_3 = 1.0$        (e) $\gamma_3 = 1.5$        (f) $\gamma_3 = 2.0$

Figure 2: Learning four data points by three-layer ReLU NNs with different $\gamma_3$'s are shown in the first row. The corresponding scatter plots of initial (red) and final (green) $\{\boldsymbol{W}_k^{[1]}\}_{k=1}^m \in \mathbb{R}^2$ are shown in the second row. $\gamma_2 = 0$, hidden neuron number m = 10000.

The first row in Fig. 2 shows typical learning results over different $\gamma_3$'s, from a relatively jagged interpolation (NTK scaling) to a smooth interpolation (mean-field scaling) and further to a linear spline interpolation. With input weight and bias, $\boldsymbol{W}_k^{[1]}$ is two-dimensional for $d = 1$. We display the $\{((\boldsymbol{W}_k^{[1]})_1, (\boldsymbol{W}_k^{[1]})_2)\}_{k=1}^m$ of the trained network in Fig. 2. For $\gamma_3 = 1.0$, the initial scatter plot is very close to the one after training. However, for $\gamma_3 = 2.0$, active neurons (i.e., neurons with significant amplitude and away from the origin) are condensed at a few orientations, which strongly deviates from the initial scatter plot. For $\gamma_3 = 1.5$, the scatter points present an intermediate state.

Inspired by $\boldsymbol{W}^{[1]}$, we found that the directions between the input weights is a very important group feature, which can also reflect the characteristics between different $\gamma_3$ and $\gamma_2$. Considering that $\boldsymbol{W}_k^{[2]}$ is $m + 1$ dimensional, We cannot directly and intuitively show $\{\boldsymbol{W}_k^{[2]}\}_{k=1}^m$ in a figure. Here, $\boldsymbol{W}_k^{[2]}$ represents the input weight of the $k$-th neuron in the second hidden layer. Therefore, to characterize the distance between different vectors in $\{\boldsymbol{W}_k^{[2]}\}_{k=1}^m$, we introduce $D(\boldsymbol{u}, \boldsymbol{v})$ to denote the inner product of two $\{\boldsymbol{W}_k^{[2]}\}_{k=1}^m$, as Eq. (2). As shown in Fig. 3, for $\gamma_3 = 2.2$, $\boldsymbol{W}_k^{[2]}$ hardly has a strong relationship with each other, while for $\gamma_3 = 2.8$, most of the $\boldsymbol{W}_k^{[2]}$ are very close to each other, showing strong condensation.

## 5.3 Phase diagram for three-layer neural networks

In this section, with $\gamma_3$ and $\gamma_2$ as coordinates, we characterize at $m \to \infty$ the dynamical regimes of NNs and identify their boundaries in the phase diagram through experimental approach. How to characterize and classify different types of training behaviors of NNs is an important open question. However, there is little work on the three-layer phase diagram or the multi-layer one. For the three-layer infinitely wide ReLU neural networks, as shown in Fig. 1, several works have proved specific points in the three-layer phase diagram belong to the linear regime of both $\boldsymbol{W}^{[1]}$ and $\boldsymbol{W}^{[2]}$. However,

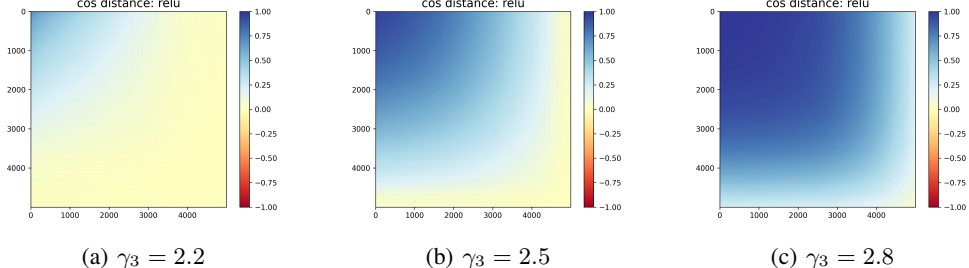

(a) $\gamma_3 = 2.2$        (b) $\gamma_3 = 2.5$        (c) $\gamma_3 = 2.8$

Figure 3: Learning four data points by three-layer ReLU NNs with different $\gamma_3$'s are shown in the first row, with $\gamma_2 = 0.5$ and hidden neuron number $m = 10000$. The color indicates $D(u, v)$ of two hidden neurons' input weights initialized by Gaussian distribution, whose indexes are indicated by the abscissa and the ordinate, respectively. If neurons are in the same navy-blue block, $D(u, v) \approx 1$ (beige block, $D(u, v) \approx 0$, and red block, $D(u, v) \approx -1$), their input weights have the same (perpendicular, opposite) direction. Considering that not all neurons are activated for ReLU activation function, we only select half of the neurons with the largest modulus length to draw $D(u, v)$.

its exact range in the phase diagram remains unclear. On the other hand, NN training dynamics can also be highly nonlinear at $m \to \infty$, such as widely used initialization method, Glorot and Bengio (2010), as a point shown in the phase diagram. However, whether there are other points in the phase diagram that has similar training behavior is not well understood in three-layer condition. In addition, it is still not clear if there are other regimes in the phase diagram that are nonlinear but behaves distinctively comparing to the Glorot and Bengio (2010) initialization method. In the following, we will address these problems and draw a three-layer phase diagram.

### 5.3.1 Regime identification and separation

To characterize different dynamical regimes, we define the relative change of weights as

$$\mathrm{RD}(\boldsymbol{W}^{[1]}) = \frac{\|\boldsymbol{\theta}^*_{\boldsymbol{W}_1} - \boldsymbol{\theta}_{\boldsymbol{W}_1}(0)\|_2}{\|\boldsymbol{\theta}_{\boldsymbol{W}_1}(0)\|_2}, \ \mathrm{RD}(\boldsymbol{W}^{[2]}) = \frac{\|\boldsymbol{\theta}^*_{\boldsymbol{W}_2} - \boldsymbol{\theta}_{\boldsymbol{W}_2}(0)\|_2}{\|\boldsymbol{\theta}_{\boldsymbol{W}_2}(0)\|_2}, \tag{10}$$

where $\boldsymbol{\theta}^*_{\boldsymbol{W}_1}$ and $\boldsymbol{\theta}^*_{\boldsymbol{W}_2}$ are the weights after training for the first and the second hidden layer, and $\boldsymbol{\theta}_{\boldsymbol{W}_1}(0)$ and $\boldsymbol{\theta}_{\boldsymbol{W}_2}(0)$ for initial weights.

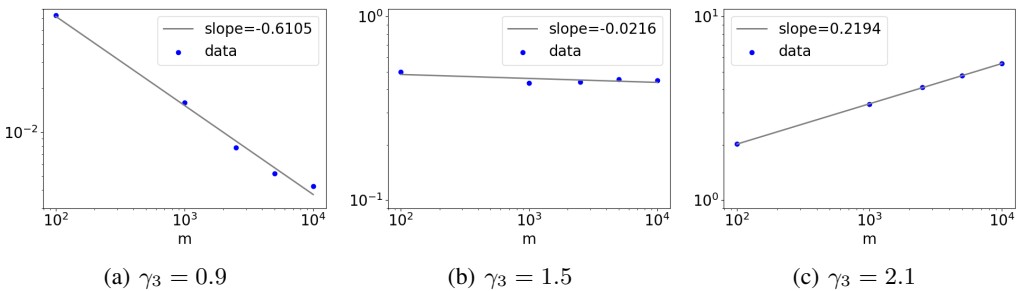

(a) $\gamma_3 = 0.9$        (b) $\gamma_3 = 1.5$        (c) $\gamma_3 = 2.1$

Figure 4: $\mathrm{RD}(\boldsymbol{W}^{[1]})$ v.s. $m$. Still learn four data points as in Fig. 2 by three-layer ReLU NNs with different $\gamma_3$'s and $\gamma_2 = 0$. The slopes in (a)–(c) are fitted by $\mathrm{RD}(\boldsymbol{W}^{[1]})$ w.r.t. $m = 100, 1000, 2000, 5000, 10000$ in a log-log scale. As $\gamma_3$ grows from 0.9 to 2.1, the corresponding slopes grow as well.

We quantify the growth of $\mathrm{RD}(\boldsymbol{W}^{[i]})$, $i = 1, 2$ as $m \to \infty$. As shown in Fig. 4(a–c), they approximately have a power-law relation. Therefore, similar to Luo et al. (2021), we define

Table 2: Two groups of $S_{W_1}$ and $S_{W_2}$. Within a group, $\gamma_2$ and $\gamma_3$ are the same, while $\alpha$, $\boldsymbol{a}$, $\boldsymbol{W}^{[2]}$ and $\boldsymbol{W}^{[1]}$ are different. These values are the average of eight experiments.

| Group | $\alpha$ | $\boldsymbol{a}$ | $\boldsymbol{W}^{[2]}$ | $\boldsymbol{W}^{[1]}$ | $\gamma_2$ | $\gamma_3$ | $S_{W_1}$ | $S_{W_2}$ | Std($S_{W_1}$) | Std($S_{W_2}$) |
|---|---|---|---|---|---|---|---|---|---|---|
| | $m^{-0.5}$ | $m^{-8/15}$ | $m^{-8/15}$ | $m^{-8/15}$ | 0 | 1.1 | $-0.4108$ | $-0.9119$ | | |
| No. 1 | 1 | $m^{-11/30}$ | $m^{-11/30}$ | $m^{-11/30}$ | 0 | 1.1 | $-0.4084$ | $-0.9364$ | $6.44\times10^{-3}$ | $1.05\times10^{-2}$ |
| | $m^{0.5}$ | $m^{-0.2}$ | $m^{-0.2}$ | $m^{-0.2}$ | 0 | 1.1 | $-0.4231$ | $-0.9310$ | | |
| | $m^{-0.3}$ | $m^{-7/6}$ | $m^{-7/6}$ | $m^{-7/15}$ | 0.7 | 2.5 | $-0.3130$ | $0.0251$ | | |
| No. 2 | 1 | $m^{-16/15}$ | $m^{-16/15}$ | $m^{-11/30}$ | 0.7 | 2.5 | $-0.3176$ | $0.0263$ | $4.64\times10^{-3}$ | $2.50\times10^{-3}$ |
| | $m^{0.3}$ | $m^{-29/30}$ | $m^{-29/30}$ | $m^{-4/15}$ | 0.7 | 2.5 | $-0.3243$ | $0.0205$ | | |

$$S_{\boldsymbol{W}_{[i]}} = \lim_{m\to\infty} \frac{\log \mathrm{RD}(\boldsymbol{W}^{[i]})}{\log m}, \tag{11}$$

which is empirically obtained by estimating the slope in the log-log plot. As shown in Table 2, NNs with the same pair of $\gamma_2$ and $\gamma_3$ , but different $\alpha, \beta_1, \beta_2$ and $\beta_3$, have very similar $S_{W_i}$, which validates the effectiveness of the normalized model. In the following experiments, we only show result of one combination of $\alpha, \beta_1, \beta_2$ and $\beta_3$ for a pair of $\gamma_2$ and $\gamma_3$.

Then, we explore the phase diagram by experimentally scanning $S_{W_i}$ over the phase space. The result for the same 1-d problem as in Fig. 2 is presented in Fig. 5. In the red zone, where $S_{W_i}$ is less than zero, $\mathrm{RD}(\boldsymbol{W}^{[i]}) \to 0$ as $m \to \infty$, indicating a linear regime. In contrast, in the blue zone, where $S_{W_i}$ is greater than zero, $\mathrm{RD}(\boldsymbol{W}^{[i]}) \to \infty$ as $m \to \infty$, indicating a highly nonlinear behavior. Their boundary are experimentally identified through interpolation indicated by stars in Fig. 5, where $\mathrm{RD}(\boldsymbol{W}^{[i]}) \sim O(1)$. The dashed lines are our speculative boundary, i.e., critical regime, by a large amount of experiments . In order to verify the rationality of the three-layer phase diagram, we also verify the results on the MNIST dataset, as shown in Fig. 6, and obtain similar results.

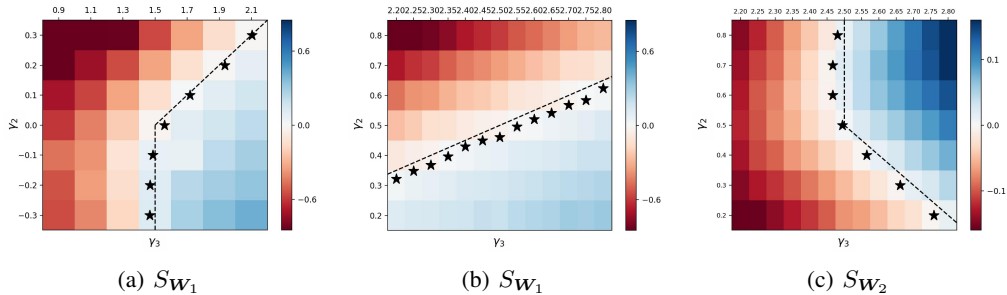

(a) $S_{W_1}$      (b) $S_{W_1}$      (c) $S_{W_2}$

Figure 5: For synthetic data, $S_{W_i}$ estimated on three-layer ReLU NNs of $m = 100, 1000, 2500, 5000$ and $10000$ hidden neurons of each layer over $\gamma_3$ (ordinate) and $\gamma_2$ (abscissa). The stars are zero points obtained by the linear interpolation over different different $\gamma_3$ for each fixed $\gamma_2$ in (a) to (c), while the stars are zero points obtained by the linear interpolation over different different $\gamma_2$ for each fixed $\gamma_3$ in (b). Dashed lines are auxiliary lines indicating our conjecture.

## 5.4 Condensation

We then examine the condensation over the phase diagram. For $\boldsymbol{W}_k^{[1]}$, we display the $\{((\boldsymbol{W}_k^{[1]})_1, (\boldsymbol{W}_k^{[1]})_2)\}_{k=1}^m$ for two different regions in Fig. 7. As is shown in Fig. 7, on the left side of the blue squares, the parameters after training and the initialization parameters are very

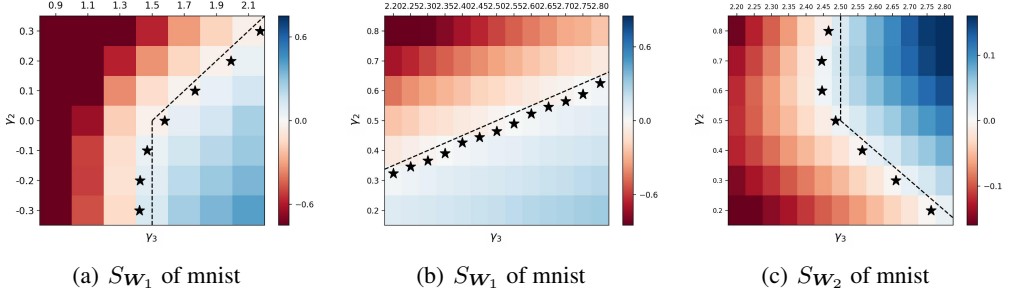

(a) $S_{\boldsymbol{W}_1}$ of mnist  (b) $S_{\boldsymbol{W}_1}$ of mnist  (c) $S_{\boldsymbol{W}_2}$ of mnist

Figure 6: For mnist data, $S_{\boldsymbol{W}_i}$ estimated on three-layer ReLU NNs of $m = 100, 1000, 2500, 5000$ and 10000 hidden neurons of each layer over $\gamma_3$ (ordinate) and $\gamma_2$ (abscissa). The stars are zero points obtained by the linear interpolation over different different $\gamma_3$ for each fixed $\gamma_2$ in (a) and (c), while the stars are zero points obtained by the linear interpolation over different different $\gamma_2$ for each fixed $\gamma_3$ in (b). Dashed lines are auxiliary lines indicating our jecture.

close, while on the right side of the blue squares, weights with not too small magnitude are condensed at a few orientations, which strongly deviates from the initial scatter plot. The graphs in the blue boxes present an intermediate state or process state, making the above-mentioned change a continuous process. Next, we need to define a quantity to describe this phenomenon.

For the second hidden layer, the weights is very high-dimensional. We define a quantity to qualitatively measure the degree of condensation. We define $\zeta = \frac{1}{(m/2)^2} \sum_{i,j=1}^{m/2} |D(\boldsymbol{u}_i, \boldsymbol{u}_j)|$, which is the average inner product distance between different $\boldsymbol{W}_{[k]}^{[2]}$'s and subscript $[k]$ indicates that the row of $\boldsymbol{W}^{[2]}$ which has a magnitude with ranking $k$ among all rows. More condensation yields to a larger $\zeta$. As is shown in Fig. 8, we could clearly observe that the condensation tendency is consistent with phase distribution in Fig. 8.

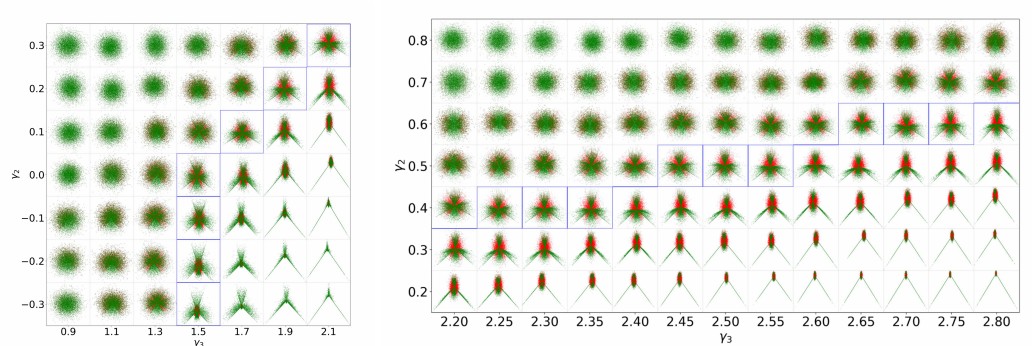

Figure 7: Regime characterization of three-layer Neural Network for $\{\boldsymbol{W}_k^{[1]}\}_{k=1}^m$ at width 10000. The network structure and the four points of the objective function are consistent with Fig. 2. $\gamma_3$ and $\gamma_2$ are indicated by the abscissa and the ordinate, respectively. The red dots in the small figure correspond to the initialization parameters, and the green dots correspond to the post-training parameters.

## 6  Conclusion

In this paper, we empirically characterized the linear, critical, and condensed regimes with distinctive features and draw the phase diagram for the three-layer ReLU NN at the infinite-width limit. The homogeneity of ReLU is is an important basis for deriving the coordinate of the phase diagram, so it could not be replaced by any non-linearity activation function. Through experiments, we further identify the condensation as the signature behavior in the condensed regime of strong non-linearity.

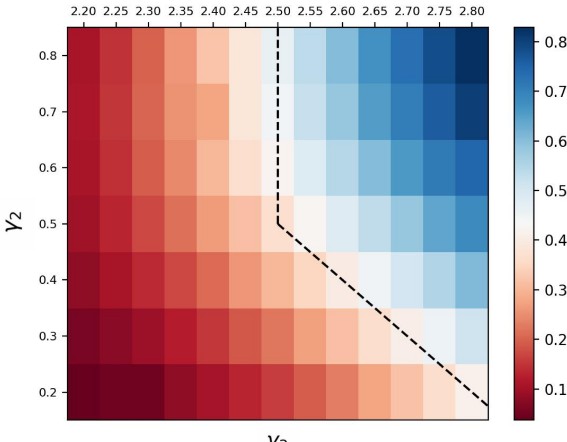

Figure 8: $\zeta$ of three-layer Neural Network for $\{\boldsymbol{W}_k^{[2]}\}_{k=1}^m$ at width 10000. The network structure and the four points of the objective function are consistent with Fig. 3. $\gamma_3$ and $\gamma_2$ are indicated by the abscissa and the ordinate, respectively. For the color of the squares in the picture, dark blue means that the $\boldsymbol{W}_k^{[2]}$'s condense in one direction, while dark red means that $\boldsymbol{W}_k^{[2]}$'s are irrelevant with each other. Since the experimental results will change with the initialization by Gaussian distribution, in order to ensure the stability and repeatability of the experimental results, we repeated the experiment for 8 times, averaged the results, and finally obtained the figure.

Our phase diagram figures out the relation between the training dynamics of multi-layer neural networks and their initialization, reveals the possibility of having different training dynamics for different layers within a neural network, and serves as a map that provide a guidance for the future study of deep NNs, such as how different dynamical regimes of different hidden layers affect the generalization.

## Acknowledgments and Disclosure of Funding

This work is sponsored by the National Key R&D Program of China Grant No. 2022YFA1008200 (Z. X., T. L., Y. Z.), the Shanghai Sailing Program, the Natural Science Foundation of Shanghai Grant No. 20ZR1429000 (Z. X.), the National Natural Science Foundation of China Grant No. 62002221 (Z. X.), the National Natural Science Foundation of China Grant No. 12101401 (T. L.), Shanghai Municipal Science and Technology Key Project No. 22JC1401500 (T. L.), the National Natural Science Foundation of China Grant No. 12101402 (Y. Z.), Shanghai Municipal of Science and Technology Project Grant No. 20JC1419500 (Y.Z.), the Lingang Laboratory Grant No.LG-QS-202202-08 (Y.Z.), Shanghai Municipal of Science and Technology Major Project No. 2021SHZDZX0102, and the HPC of School of Mathematical Sciences and the Student Innovation Center, and the Siyuan-1 cluster supported by the Center for High Performance Computing at Shanghai Jiao Tong University, Key Laboratory of Marine Intelligent Equipment and System, Ministry of Education, P.R. China.

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
