# OpenReview forum: "Empirical Phase Diagram for Three-layer Neural Networks with Infinite Width"
_NeurIPS.cc/2022/Conference — NeurIPS 2022 Accept_

### Official Review · Reviewer_iryP · 2022-07-10

**Rating:** 6
**Confidence:** 3
**Soundness:** 3 good
**Presentation:** 4 excellent
**Contribution:** 3 good

**Summary:**

This work studies empirically the effect of initialization scalings on the dynamics of three layer neural networks. The main result is a phase diagram exhibiting quite clearly the presence of different dynamical regimes for the two different layers (condensed and linear).

**Questions:**

- Is the phase diagram of Fig. 1 plotted based on theoretical or empirical lines ? This should be made more clear.
- « For instance, if we want to study the nonlinear effect of a hidden layer, we can set this interested hidden layer in the non-linear regime and all other hidden layers in the linear regime » what do the authors mean by « study the nonlinear effect » ? This is vague and does not match the description of « guidance for hyper parameter tuning » mentioned in the sentence before
- «  the dynamics of a is not the main concern » : could the authors provide more justification on this ?
- I do not understand the following sentence (« which leads to the initialization parameters... »), and the parameter B in beta2 = B beta3 does not seem to be defined or used anywhere
- Figures 5 and 6 : the caption mentions that the width m is varied across 5 different values but I am confused about how that reflects in the figures : are the results averaged over the different values of m ?
- It would be nice to have a discussion on the impact of the nonlinearity. For example including a phase diagram for Tanh ?

**Limitations:**

Yes

**Strengths And Weaknesses:**

## Strengths

- Despite a few typos the paper is well presented and easy to read.
- The phase diagram of figure 1 is neat and a valuable contribution in my opinion. It is well illustrated by the empirical figures of Section 5.
- The empirical investigation is thorough.

## Weaknesses

- The most important part of the paper, where the parameters gamma2 and gamma3 are derived, is rushed and difficult to understand. The authors should clarify their choice as it looks a bit like dark magic right now
- Some important works on phase diagrams are missing in related works, for example the work on disentangling feature and lazy regime by Mario Geiger and colleagues and the series of papers on tensor programs and muparametrization by Greg Yang and colleagues.

---

> ### Author Response · Authors · 2022-08-02
> **Response to Reviewer iryP**
>
> $\mathrm{Point 1: }$
>
> The most important part of the paper, where the parameters gamma2 and gamma3 are derived, is rushed and difficult to understand. The authors should clarify their choice as it looks a bit like dark magic right now.
>
> $\mathrm{Reply: }$
>
> First, we compute the dynamical system of three-layer neural networks, and then unitize the parameters of each layer. Combined with the homogeneity of $\mathrm{ReLU}$ function, we finally replace the unitized parameters in $\kappa_1, \kappa_2, \kappa_3$, shown by Eqs. 5,6,7,8. Thanks for your suggestion, and we now make it clearer in the article.
>
> $\mathrm{Point 2: }$
>
> Some important works on phase diagrams are missing in related works, for example the work on disentangling feature and lazy regime by Mario Geiger and colleagues and the series of papers on tensor programs and mu parametrization by Greg Yang and colleagues.
>
> $\mathrm{Reply: }$
>
> Thanks. We have added the recommended references to the ‘Related works’ in line 89 now.
>
> $\mathrm{Point 3: }$
>
> Is the phase diagram of Fig. 1 plotted based on theoretical or empirical lines? This should be made more clear.
>
> $\mathrm{Reply: }$
>
> Fig.1 is plotted based on empirical lines. Now, we make it clear in the captions.
>
> $\mathrm{Point 4: }$
>
> « For instance, if we want to study the nonlinear effect of a hidden layer, we can set this interesting hidden layer in the non-linear regime and all other hidden layers in the linear regime » What do the authors mean by « study the nonlinear effect » ? This is vague and does not match the description of « guidance for hyper parameter tuning » mentioned in the sentence before.
>
> $\mathrm{Reply: }$
>
> For example, if we want to study how the regime of the first hidden layer affects the generalization, we can compare two regimes: (a) whose first hidden layer is ''condensed'' (a.k.a. ''non-linear'') and the second hidden layer is ''linear'', and (b) whose two hidden layers are both ''linear''. Here, we make the second hidden layer ''linear'' in the comparative study. (Of course, you can also make the second hidden layer ''condensed'' in the comparative study.) Since we now have a clear phase diagram, which is like a map, guiding us towards hyper-parameter tuning, these experiments would be easy to carry out.
>
> $\mathrm{Point 5: }$
>
> «  the dynamics of a is not the main concern »: could the authors provide more justification on this?
>
> $\mathrm{Reply: }$
>
> $a$ is the input weight of the output layer. Since there is no activation for the output layer, it is always linear of the output with respect to $a$.
>
> $\mathrm{Point 6: }$
>
> I do not understand the following sentence (« which leads to the initialization parameters... »), and the parameter B in beta2 = B beta3 does not seem to be defined or used anywhere
>
> $\mathrm{Reply: }$
>
> We reformulate this part by
> ``Considering that we are mainly concerned with the dynamics of $W^{[1]}$ and $W^{[2]}$, i.e., the dynamics of $a$, which is linear with respect to the output, is not the main concern, and the initialization methods used for a neural network in practice often has $\beta_2 = B \beta_3$. So, we have the following assumption."
>
> By this assumption, we have $\kappa_1=B$ and $\gamma_1=0$ when m goes to infinity. So we simplify our discussion on the phase diagram of three layer network under the coordinates: $(\gamma_1, \gamma_2, \gamma_3)= (0, \gamma_2, \gamma_3)$.
>
> $\mathrm{Point 7: }$
>
> Figures 5 and 6: the caption mentions that the width m is varied across five different values, but I am confused about how that reflects in the figures: are the results averaged over the different values of m?
>
> $\mathrm{Reply: }$
>
> $S_{W_{i}}$  in figures is an estimated slope for the relation between $\mathrm{RD}(W^{[i]})$ and $m$. We choose $m = 100,1000,2000,5000,10000$ and corresponding $\mathrm{RD}(W^{[i]})$ to calculate $S_{W_{i}}$.
>
> $\mathrm{Point 8: }$
>
> It would be nice to have a discussion on the impact of the non-linearity. For example, including a phase diagram for Tanh?
>
> $\mathrm{Reply: }$
>
> The homogeneity of ReLU is important to derive the coordinate for the phase diagram. For Tanh, this homogeneity disappears. It is not a simple extension from ReLU to any non-linearity and we are working on it.

---

> > ### Comment · Reviewer_iryP · 2022-08-03
> > **Response to rebuttal**
> >
> > Thanks for your replies !
> >
> > “For example, if we want to study how the regime of the first hidden layer affects the generalization, we can compare two regimes: (a) whose first hidden layer is ''condensed'' (a.k.a. ''non-linear'') and the second hidden layer is ''linear'', and (b) whose two hidden layers are both ''linear’’”
> > What I mean is that nobody is interested in the linear regime in practice, so it would be better not to mention this as guidance for hyper parameter tuning in practice, but rather as a tool for the theory community.
> >
> > “The homogeneity of ReLU is important to derive the coordinate for the phase diagram. For Tanh, this homogeneity disappears. It is not a simple extension from ReLU to any non-linearity and we are working on it.”
> > I see. This should be stated much more clearly in the limitations section !

---

> > > ### Author Response · Authors · 2022-08-05
> > > **Response to Reviewer iryP**
> > >
> > > Dear Reviewer iryP
> > >
> > > Thanks for your response and suggestion!
> > >
> > > We now replace < ... guidance for tuning parameters in practical training. > by < ... a tool for the theory community. > and will state the limitations clearly!
> > >
> > > Best
> > >
> > > Authors.

---

### Official Review · Reviewer_1ShU · 2022-07-12

**Rating:** 7
**Confidence:** 4
**Soundness:** 4 excellent
**Presentation:** 4 excellent
**Contribution:** 3 good

**Summary:**


Building on Luo et al. 2021, the current submission presents a detailed numerical study of gradient flow of feedforward neural networks with different scaling parameters used for their initialization.

The experiments reveal that different layers of the three-layer architecture exhibit  changes to their parameters that are:

1. negligible, aka *linear* regime

2. constant wrt. to network size aka *critical* regime

3. increasing with network size, aka *condensed* regime.

Analogously to Luo et al. 2021, the above regimes are revealed to be codimension two bifurcations, controlled by quantities $\gamma_2, \, \gamma_3$ defined in Eq. 9.

By carefully varying $\gamma_2, \, \gamma_3$ for each hidden layer, the authors demonstrate that the weights can evolve separately and in tandem in either of the above three regimes.

**Questions:**

Use of cosine similarity:
Am I correct to assume that the cosine similarity is specific to ReLU is positive homogeneous?

**Strengths And Weaknesses:**

To the best of my knowledge, this submission gives new insight into training dynamics of MLPs with more than one hidden layer.
The numerical nature of the study is well complemented by its clever analysis of the bifurcation parameters.
This in turn allows for an elegant and insightful analysis of what I believe is considered a dauntingly difficult problem.

I have not kept up to date with the most recent analyses of the gradient dynamics of shallow NNs, and hence I will defer to other reviewers for estimates of the impact of this work.
Notwithstanding that, I greatly value the elegance and insights that this submission presents.

Minor comments:

1.  The manuscript would benefit from proofreading, especially in regards to small grammatical and stylistic errors.

2. Figure 3: Perhaps a diverging colormap would be better suited for $D(u,v)$?

3. Table 2 and Figure 4: I would like to encourage the authors to report some measure of spread for the experimentally collected quantities.

4. Figures 4 & 5: please label the axis with $\gamma_2, \gamma_3$. I believe the captions indicating the ordinate and abscissa are inconsistent with numerical ranges explored elsewhere, e.g. in figure 6.

5. Figure 3 and 8: mention or indicate graphical the expectation for $D(u,v)$ and $\zeta$ under a Gaussian distribution over the weights

---

> ### Author Response · Authors · 2022-08-02
> **Response to Reviewer 1ShU**
>
> $\mathrm{Point 1: }$
>
> The manuscript would benefit from proofreading, especially in regards to small grammatical and stylistic errors.
>
> $\mathrm{Reply: }$
>
> We thank the reviewer for your thoughtfulness and insightful comments on the quality of writing/presentation. We will double-check and revise the problems about writing.
>
> $\mathrm{Point 2: }$
>
> Figure 3: Perhaps a diverging colormap would be better suited for $D(u,v)$
>
> $\mathrm{Reply: }$
>
> Thank for your advice, and we now redraw it in more distinguishable colors.
>
> $\mathrm{Point 3: }$
>
> Table 2 and Figure 4: I would like to encourage the authors to report some measure of spread for the experimentally collected quantities.
>
> $\mathrm{Reply: }$
>
> Thank for your advice, and we now add the standard deviations of $S_{W_1}$ and $S_{W_2}$ to Table 2. As for Figure 4, the number of neurons of neural networks used is relatively large, so the fluctuation of $\mathrm{RD}(W^{[1]})$ is relatively small. Besides, we have also repeated the experiments several times to gain the value of $\mathrm{RD}(W^{[1]})$, so we do not provide further details. Thank for your advice, and we will report some measures of spread in our revision.
>
> $\mathrm{Point 4: }$
>
> Figures 4 & 5: please label the axis with $\gamma_2$, $\gamma_3$. I believe the captions indicating the ordinate and abscissa are inconsistent with numerical ranges explored elsewhere, e.g. in figure 6.
>
> $\mathrm{Reply: }$
>
> Thank for your advice, and we have labeled the axis with $\gamma_2$ and $\gamma_3$ for Fig. 5, Fig. 6 and Fig.8 in the article.
>
> $\mathrm{Point 5: }$
>
> Figure 3 and 8: mention or indicate graphical the expectation for $D(u, v)$ and $\zeta$ under a Gaussian distribution over the weights
>
> $\mathrm{Reply: }$
>
> Thank for your advice, and now we mention the expectation under a Gaussian distribution over the weights in the caption of Fig. 3 and Fig. 8.
>
> $\mathrm{Point 6: }$
>
> Use of cosine similarity: Am I correct to assume that the cosine similarity is specific to ReLU is positive homogeneous?
>
> $\mathrm{Reply: }$
>
> Yes.

---

### Official Review · Reviewer_PDY9 · 2022-07-15

**Rating:** 4
**Confidence:** 3
**Soundness:** 2 fair
**Presentation:** 1 poor
**Contribution:** 1 poor

**Summary:**

This paper builds upon the work of Luo et al., 2021 and characterizes the phase diagram for different dynamical regimes in the case of a three-layer neural network at infinite width. Based on appropriately chosen quantities, the paper shows that different layers can be simultaneously in the linear, critical or condensed regime. The proposed phase diagram is backed with several experiments that depict the nature of the learned function and dynamics in these regimes.

**Questions:**

*Firstly, check out the first point under weaknesses. I would be very interested in looking at additional results related to the mentioned questions therein.*

Besides, there are also some other questions and suggestions that I have related to the experiments:

- Figure 3: Can you make the plot for the similar setting as in Figure 2? It is better to have results in comparable settings. Also, do you see more condensation in the initial layers or towards the output?

- How do the results in Figure 2 and 3 with increasing number of samples? Do we see that some initialization regimes coalesce when number of samples is large?

- Can you share a similar plot as in Figure 4 for RD(W^[2])?

**Limitations:**

NA.

**Strengths And Weaknesses:**

## Strengths
- I personally find this line of work of characterizing the phase diagram to be quite insightful. It provides a useful perspective to think about neural networks.

## Weaknesses:
- The phase diagram partitions the space into four possibilities based on the choice of regime for each layer. However,  more or less all the experiments and results are in the setting when the second hidden layer is in the linear regime. As a result, most of their findings are qualitatively very similar to that observed in Luo et al., 2021. Given their richer characterization of the phase diagram as compared to prior work, the authors nevertheless do not discuss these potential implications: e.g., it would have been understanding to compare what happens in the condensed regime for both hidden layers. As well as, how does condensed regime in a deeper layer while being linear in the lower layer compare to the converse scenario? Further, quantitatively exploring the advantages/disadvantages of being in these different regimes on generalization/optimization could have been worthwhile too. So, I think there are several missed opportunities here and the paper is not properly fleshed out, and overall lacks in contribution.
- On this note, this work is heavily inspired and based on the Luo et al., 2021 -- which in itself is not wrong, but their manner of presentation is misleading. For example, the quantities gamma_3, gamma_2, kappa_3 are essentially identical to the gamma, gamma^\prime, kappa in Luo et al 2021 (gamma_3 is just 1/2 more than gamma since here we have one additional layer). Likewise most of the considered experiments are modeled based on this prior work, and even most of the presented results/figures are quite similar. Again to emphasize, it is absolutely alright to build up on results and experiments from past work, but there should be appropriate acknowledgement --- which are missing here. To give another example, the authors write in section 5.3.1 of regime identification and separation, that "we define the relative change of weights... " but it is the same as Luo et al 2021 and even the results in the figure below are basically the same as in that work.

### Minor Remarks:
- For Xavier initialization, the values of the gamma_2 and gamma_3 should be interchanged. Also, in the same row for Xavier init, the the value of kappa_3 seems wrong.
- Writing is a bit sloppy and redundant at times: some lines are repeated multiple times, in the abstract, at the beginning of introduction, and towards the end of introduction. And often there are many grammatical mistakes, so it would be a good idea to fix them as well.
- The figures do not come out well in print.

---

> ### Author Response · Authors · 2022-08-02
> **Reponse to Reviewer PDY9**
>
> $\mathrm{Point 1: }$
>
> However, more or less all the experiments and results are in the setting when the second hidden layer is in the linear regime. As a result, most of their findings are qualitatively very similar to that observed in Luo et al., 2021.
>
> $\mathrm{Reply: }$
>
> We want to clarify a misunderstanding, i.e., our work does not restrict the second hidden layer in the linear regime.
>
> We make an assumption $\beta_3=B\beta_2$, which leads to $\kappa_1=0$ as $m$ goes to infinity. This assumption is reasonable since commonly used initializations in Table. 1 follow the above assumption. So we here discuss all the possible cases when the coordinate is set as $(0, \gamma_2, \gamma_3)$. As shown in Fig. 1, in the purple zone, both two hidden layers are in the condensed regime, while in the cream zone, both two hidden layers are in the linear regime. In the green zone, the first hidden layer is in linear regime, with the second hidden layer in condensed regime, while in the blue zone, the opposite one holds.
> Therefore, the second hidden layer is not always in linear regime, which can also be seen in Fig. 5(c).
>
> $\mathrm{Point 2: }$
>
> Given their richer characterization of the phase diagram as compared to prior work, the authors nevertheless do not discuss these potential implications: e.g., it would have been understanding to compare what happens in the condensed regime for both hidden layers. As well as, how does condensed regime in a deeper layer while being linear in the lower layer compare to the converse scenario? Further, quantitatively exploring the advantages/disadvantages of being in these different regimes on generalization/optimization could have been worthwhile too. So, I think there are several missed opportunities here and the paper is not properly fleshed out, and overall lacks in contribution.
>
> $\mathrm{Reply: }$
>
> We agree with the reviewer's comments. But it is really hard to discuss all the issues in a conference paper. For example, [1] introduces the Neural Tangent Kernel, leaving many properties unexplored and playing an important role for deep learning study. In this paper, we mainly present some significant results on the relation between neural network dynamics and hyper-parameters, which is helpful to the study of the characteristics of different regimes, such as optimization, generalization, implicit regularization, etc. These are also our ongoing work.
>
> [1] https://arxiv.org/abs/1806.07572
>
> $\mathrm{Point 3: }$
>
> For example, the quantities gamma_3, gamma_2, kappa_3 are essentially identical to the gamma, gamma$^{\prime}$, kappa in Luo et al. 2021 (gamma_3 is just 1/2 more than gamma since here we have one additional layer).
>
> $\mathrm{Reply: }$
>
> Although the quantities $\gamma_3, \gamma_2$, $\kappa_3$, in the three-layer phase diagram, are similar to the $\gamma, \gamma^{\prime}$, $\kappa$ in Luo et al. 2021, more hidden layers make the study of neural networks more complicated. By introducing $\beta_3$ and $\beta_2$ and appropriately constructing $\gamma_3, \gamma_2$, $\kappa_3$, the whole derivation is now simplified and the three-layer phase diagram is clearly displayed. We would comment on this similarity and cite Luo et al., 2021 properly in the revision.
>
> $\mathrm{Point 4: }$
>
> Likewise, most of the considered experiments are modeled based on this prior work, and even most of the presented results/figures are quite similar. Again to emphasize, it is absolutely alright to build up on results and experiments from past work, but there should be appropriate acknowledgement --- which are missing here.
>
> $\mathrm{Reply: }$
>
> Thanks for your suggestion, we will give proper credit to Luo et al. 2021.
>
> $\mathrm{Point 5: }$
>
> To give another example, the authors write in section 5.3.1 of regime identification and separation, that "we define the relative change of weights... " but it is the same as Luo et al. 2021 and even the results in the figure below are basically the same as in that work.
>
> $\mathrm{Reply: }$
>
> Thanks for your suggestion, we have cited Luo et al. 2021 and revised this in the text.
>
> $\mathrm{Point 6: }$
>
> For Xavier initialization, the values of the gamma_2 and gamma_3 should be interchanged. Also, in the same row for Xavier init, the value of kappa_3 seems wrong.
>
> $\mathrm{Reply: }$
>
> Thanks for your suggestion, we have corrected Table 1 and made sure the values are accurate.

---

> ### Author Response · Authors · 2022-08-02
> **Reponse to Reviewer PDY9**
>
> $\mathrm{Point 7: }$
>
> Writing is a bit sloppy and redundant at times: some lines are repeated multiple times, in the abstract, at the beginning of introduction, and towards the end of introduction. And often there are many grammatical mistakes, so it would be a good idea to fix them as well.
>
> $\mathrm{Relpy: }$
>
> Thanks for your suggestion, we have double-checked the paper for grammatical and stylistic errors, making it easier to follow.
>
> $\mathrm{Point 8: }$
>
> The figures do not come out well in print.
>
> $\mathrm{Relpy: }$
>
> Thanks for your suggestion, we now adjust our figures to better demonstrate them.
>
> $\mathrm{Point 9: }$
>
> Figure 3: Can you make the plot for a similar setting as in Figure 2? It is better to have results in comparable settings. Also, do you see more condensation in the initial layers or towards the output?
>
> $\mathrm{Relpy: }$
>
> Yes. However, we do not plot them in the same setting due to the following reasons. The corresponding condensed zones of different layers are different, as shown in Fig.1. We want to show distinct behaviors of the second hidden layer in Fig. 3. If we use the same settings in Fig. 2, we would not see such distinct phenomena.
>
> We do not see more condensation in any layer. The condensation regime for each layer is different. It is technically hard to compare the degree of condensation of different layers.
>
> $\mathrm{Point 10: }$
>
> How do the results in Figure 2 and 3 with increasing number of samples? Do we see that some initialization regimes coalesce when the number of samples is large?
>
> $\mathrm{Relpy: }$
>
> No. The regime is only determined by hyper-parameters.
>
> $\mathrm{Point 11: }$
>
> Can you share a similar plot as in Figure 4 for $RD(W^{[2]})$?
>
> $\mathrm{Relpy: }$
>
> Yes, we will add a similar plot in Appendix.

---

> ### Author Response · Authors · 2022-08-09
> **Looking forward to hearing more**
>
> Dear Reviewer,
>    Thanks for your comments, and we are looking forward to hearing more from you on our response.
> Best,
> Authors.

---

### Meta-Review · Area_Chair_7UYa · 2022-08-27

**Recommendation:** Accept
**Confidence:** Less certain

**Metareview:**

This work studies the effect of initialization scaling on the gradient flow training dynamics of three layer (two hidden layer) MLPs. . Extending Luo et al. 2021's analysis, the authors identify "linear", "critical" and "condensed" regime depending on initialization scaling parameters

Strength pointed out by the authors include, "new insight into training dynamics of MLPS with more than one hidden layer", "elegant and insightful analysis" for a "dauntingly difficult problem" and that "phase diagram is neat and valuable contribution"

Reviewer `PDY9` saw major weakness as treating the second layer as linear regime only, which makes the analysis similar to previous work (Luo et al., 2021). In general, the reviewer believes that the full comprehensive analysis on different phases for each layer is lacking. Although the reviewer did not respond to the AC, this issue has been addressed by author response. Also please do follow the recommendation of the reviewers in terms of improving writing (better description of gamma_2, gamma_3 for example), improving related works, as well as making figures more legible.

Overall, while the paper is somewhat borderline, there are interesting insights and analysis as the reviewers pointed out without critical issues. I recommend accepting this work at NeurIPS 2022.


**Award:**

No

---

### Decision · Program_Chairs · 2022-09-14

Accept